# The Relationship between Psychological Safety and Management Team Effectiveness: The Mediating Role of Behavioral Integration

**DOI:** 10.3390/ijerph20010406

**Published:** 2022-12-27

**Authors:** Emil Viduranga Mogård, Ole Bendik Rørstad, Henning Bang

**Affiliations:** Department of Psychology, University of Oslo, 0317 Oslo, Norway

**Keywords:** team performance, management teams, psychological safety, team effectiveness, team outcomes, individual satisfaction, task performance

## Abstract

This study explores whether there is an indirect effect of psychological safety on team effectiveness in management teams, operating through the mediating variable of behavioral integration. Whilst there exists a fair amount of research on the relationship between psychological safety and team effectiveness, few have looked at potential mechanisms that can explain this association in management teams. We propose behavioral integration to be a potential mediator. Data are collected from 1150 leaders in 160 Norwegian management teams, answering a questionnaire measuring team functioning and effectiveness. Team size ranged from 3 to 19 members. Our results show a significant indirect effect of psychological safety on management team effectiveness, mediated by behavioral integration. Thus, the more team members perceive the climate as safe in terms of speaking their mind without the fear of repercussions, the more they partake in mutual collaboration, information sharing and experience ownership in the decisions being made. This is associated with management teams performing better. We also found a positive relationship between psychological safety and behavioral integration, and—in line with previous studies—that psychological safety and behavioral integration both were positively related to team effectiveness. This study adds to the existing team research literature by expanding our knowledge about the importance of psychological safety and the way it influences management team performance at all levels throughout the organizational hierarchy.

## 1. Introduction

In the summer of 2018, while fans around the globe gathered to watch the highly anticipated World Cup soccer tournament in Russia, they suddenly turned their attention to Thailand. An unprecedented rescue mission was unfolding in a remote cave complex near the borderlands of Myanmar. Six miles, deep within the mountain range that separates Thailand and its neighboring country in the north, lies the Tham Luang—the fourth biggest Thai cave system. Days had passed since last seeing the twelve boys from the Wild Boars’ soccer team and their coach, believed to have ventured into the cave post practice on 23 June [1]. The torrential rainfalls, which had flooded the cave and the discovery of shoes and bikes near the entrance first raised the alarm. Internationally renowned cave divers were called to locate the boys who had been untraceable for nine days, all whilst battling poor visibility, strong currents and complex cave systems. On 2 July, against all odds, they made contact. Through narrow passageways, murky waters and 2.5 miles from the cave entrance, stranded on a ledge, were the missing 13 people. The boys had survived.

What followed was a masterclass in international teamwork. Complementary contributions from experts around the world was given in a range of areas, including child nutrition, logistics, radio technology and weather forecast modelling. All of which resulted in a most impressive rescue operation, initially deemed impossible. In the aftermath, several people tried to make out the inner workings of the daunting operation [2] and the teams participating in it, leading to one professor at The Wharton School, University of Pennsylvania calling it an exemplary lesson of leadership [3].

Regardless of whether teams operate in such extreme, isolated and confined environments or in ordinary office settings, they are generally viewed as critical for success in modern organizations [4]. Much work in today’s organization is accomplished through collaboration, defined as the joint ownership of decisions and collective responsibility for outcomes [5]. The ever so intensifying competition for resources and demand for high performances are pressing firms to become more flexible, fast-paced and result-focused, which yields business problems that are more complex than any leader can solve singlehandedly [6]. In many contemporary firms, teams have become the building blocks on all levels of the organizational hierarchy [7].

One type of team holding a particularly central position in the organizational hierarchy is the management team. Most organizations use management teams at different levels, consisting of leaders from separate organizational units, assigned to set overall goals and strategies, coordinate tasks and engage in complex decision-making vital to the organization and its performance [7,8]. Indeed, in these highly competitive settings, exceedingly dealing with complex judgements and uncertainty coordinated across multiple individuals, some management teams do perform better than others. What characterizes an effective management team, and what may increase its chances of performing well?

## 2. Theory and Hypotheses

A line of research that has received significant attention for its impact on team performance and ultimately organizational outcomes is *team psychological safety*. Team psychological safety involves members’ perception of the consequences of taking interpersonal risks in specific contexts in the workplace [9,10]. More specifically, “when people have psychological safety at work, they feel comfortable sharing concerns and mistakes without fear of embarrassment or retribution” [11]. It is a state of mind related to the feelings and emotions each team member experiences while being part of a team [10]. It is argued that if members perceive the interpersonal climate within their teams as safe, they will be more willing to contribute their ideas and actions to collective work [12]. Edmondson [9] found that teams that developed team psychological safety cooperated better than those that did not. Prior research indicates that team psychological safety is a critical factor in understanding phenomena such as teamwork, team learning, communication [10], and that it is directly associated with team effectiveness and firm performance [9,13,14]. None of these studies looked specifically at performance in management teams.

In this study, we suggest that *behavioral integration* may serve as a mediator in the relationship between team psychological safety and management team effectiveness. Leading management researcher Donald C. Hambrick introduced the concept in 1994, and defined it as “the degree to which the group engages in mutual and collective interaction” [15]. He argued that a team is behaviorally integrated when its members share information, resources and decisions, and hence, work like a real team [15,16]. Studies following this line of research have found that behavioral integration is positively related to team and organizational processes and outcomes, e.g., firm performance [17], management teams’ decision quality [18] and employee work outcomes [19], and negatively related to organizational decline [18] and team affective conflicts [20].

Research conducted by Edmondson and colleagues indicates that a psychologically safe team climate can set the stage for “more challenging, more honest, more collaborative, and thus also more effective work environment” [11]. Hence, such a team climate might enable team members to engage more of themselves in a team, and make contributing actions to the collective, without fear of embarrassment or retribution.

We suggest that the level of team psychological safety may influence the team members’ willingness to share information and resources, and to feel collective responsibility for the team’s decisions, and thereby, enhance management team effectiveness.

The main aim of our study is to provide further insight into the mechanisms operating in the relationship between psychological safety and team performance, exploring the extent to which behavioral integration can serve as a mediator.

### 2.1. Clarifying the Concepts

#### 2.1.1. Management Teams

The present study was conducted on management teams, defined as a group of managers reporting to a superordinate manager, responsible for coordinating and providing direction for sub-units under their jurisdiction, and for the overall performance of a business unit [21,22,23]. Group members are interdependent and their responsibilities often involve setting goals, making decisions, prioritizing and designing strategies deemed important for firm performance.

There is an increased interest in better understanding the role of management teams and their effect on organizational outcomes. A growing body of literature has, in recent decades, signaled the importance of shared leadership; “a dynamic, interactive influence process among individuals in groups for which the objective is to lead one another to the achievement of group or organizational goals (…)” [24]. Thus, teams are critical for success in today’s organizations [4], with further research yielding conclusions that the top team rather than the top person has the greatest effect on firms’ functioning [25,26].

#### 2.1.2. Team Effectiveness

Team effectiveness is often defined as a multifaceted concept, and emphasizes criteria emerging both from the productive results of the team and outcomes pertaining more to the well-being of its members [27]. Hackman [28] offers a three-dimensional model, where a team’s effectiveness is defined as the extent to which the team meets or exceeds the expectations of others (task performance), leads to the learning and well-being of team members (individual well-being and growth) and facilitates and grows team capabilities for members to work interdependently in the future (team viability) [29,30]. Thus, an effective management team makes a positive contribution to the organization’s value creation process—either to the team’s outcome or for the individuals in it—relative to what could have been achieved with members only operating as individuals.

In line with recent researchers [8,31], we single out team viability as a special type of outcome, different from task performance, and individual well-being and growth. We view team viability as an emergent state or as a dynamic property of a team, which mainly arise and vary as a function of team context, input, processes and outcomes [32]. Hence, we do not include team viability in our conception of management team effectiveness but use task performance and individual well-being and growth (referred to here as individual satisfaction) as indicators of team effectiveness.

#### 2.1.3. Task Performance

The first indicator of team effectiveness is task performance. Task performance concerns the team’s productive output; namely, its products, services or decisions, and to what degree it meets certain expectations (e.g., standards of quality, quantity and timeliness) “perceived by those who review, receive or use the output” [28]. In this study, we adopt Bang and Midelfart’s definition of task performance; namely, to what extent “the results produced by the team make a significant and positive contribution to the success of the organization–in other words that the results create substantial added value for the organization” [8]. 

Bang and Midelfart [8] build on Drath et al. [33] and summarizes the added value an effective management team creates for an organization into three likely outcomes; *direction*—reasonably agreed upon and understood aim, vision or goal; *alignment*—coordinated processes and strategies of different organizational units; *commitment*—willingness of members to engage in mutual commitment in a collective effort. Hence, in our study, task performance refers to the added value a management team creates in their attempt to forge high quality decisions and create direction, alignment and commitment.

#### 2.1.4. Individual Satisfaction

The second indicator of team effectiveness is individual satisfaction, defined as the degree to which the team contributes to the members’ motivation, learning, development and personal well-being [28]. An effective management team can bring on added value for the individual team members in and between team meetings by, e.g., providing valuable advice and relevant information, thus, energizing and motivating members becoming better at their jobs [8].

By including both these indicators of team effectiveness, one accounts for the added value of an effective management through a personal level, as well as the overall performance of the team as perceived by its members.

#### 2.1.5. Team Psychological Safety

Revisiting the interest in psychologically safe environments first proposed by Schein and Bennis [34], Edmondson introduced the concept of team psychological safety—hereby referred to as *psychological safety* [9]. Psychological safety can be defined as a “shared belief that the team is safe for interpersonal risk taking” [9], meaning that others on the team will not embarrass, reject or punish you for speaking up with an idea, questions, concerns or for making mistakes. The level of psychological safety in a team has been linked to an array of different outcomes, from information exchange, knowledge sharing, creativity, voice behavior and learning behavior, to firm performance, organizational learning and culture change [10].

#### 2.1.6. Behavioral Integration

Hambrick [15] introduced behavioral integration as the most important process characterizing effective management teams. Behavioral integration is a three-dimensional construct that emphasizes the level of collaborative behavior, the level of informational exchange and the level of perceived joint decision-making by its team members [15]. According to Carmeli and Schaubroeck, “a behaviorally integrated TMT [top management team] is characterized by intense interaction that produces open information exchange and collaboratively based solutions and decisions. Such collectively derived decisions normally receive higher commitment and follow-up from members of the team” [18].

### 2.2. Psychological Safety

#### 2.2.1. Past Research on Psychological Safety

The fast-paced and hypercompetitive environment of today’s world have rendered continuous learning, innovation and change as crucial aspects of an organization’s ability to achieve success; thus, relying heavily on individuals’ and groups’ capacity to engage in behaviors such as speaking up, collaborating and experimenting [35,36]. This implies an underlying expectation for employees to take a more active role, which necessarily diverted researchers to explore the factors central to assisting them in doing so—namely, what fosters willingness to invest their energies in work and interpersonal risk-taking [37]. A key factor resulting from this line of research was psychological safety [9].

Going back as far as the 1940s to Kurt Lewin and his unfreeze-change-refreeze process for organizational change, and further developed in Schein [38], Lewin proposed that for change to occur, one had to break the pressure to sustain the status quo (unfreeze), change, and then refreeze to institutionalize change. Since then, there have been numerous endeavors trying to untangle the cognitive and social mechanics for groups and individuals in need to do so [39]. Kahn [37] was instrumental in taking psychological safety from a more contextual determinant of the organizational climate [34] to a more organization behavior-oriented focus, bringing attention to individual psychological safety. He defined it as “feeling able to show and employ one’s self without fear of negative consequences to self-image, status, or career” [37].

These conceptualizations do converge around a single unifying premise in that their purpose is to mitigate the perception of interpersonal risk-taking being too high. However, it was not before 1999, when Amy Edmondson introduced an empirical measure of psychological safety, that the term proliferated and gained the interest it holds today [9]. Unlike her predecessors who conceptualized the term as a contextual predictor for organizational change [34] and at the individual level [37], Edmondson introduced psychological safety as a shared belief at the team level. “Perceptions of psychological safety, like other such beliefs, should converge in a team, both because team members are subject to the same set of structural influences and because these perceptions develop out of salient shared experiences” [9].

Today, psychological safety is understood as the taken-for-granted belief of how others will respond if you engage in risky behaviors, such as voicing concerns, asking questions and proposing new ideas—ideas are generally perceived as putting oneself on the line [9]. Hence, a psychologically safe environment represents conditions where one feels comfortable in doing so. Nembhard and Edmondson refer to it as a tacit calculus we engage in at certain behavioral decisions points in which we “assess the interpersonal risk associated with at given behavior. (…) one weighs the potential action against the particular interpersonal climate, as in ‘if I do X here, will I be hurt, embarrassed or criticized?’” [36]. A sense of psychological safety stems from a negative answer to this question so that we further engage in these behaviors, which would have been less likely if the perceptions of the interpersonal climate and consequences were different.

The feeling of interpersonal risk can influence the team members’ willingness to show behaviors of collaboration and experimentation within a work environment. The easiest way out when perceiving the psychological safety climate as unsafe, is to avoid these behaviors. Although minimizing individual risk, this may lead to unique information being withheld by the individual; thus, potentially limiting a team’s performance [9,36]. In a recent extensive study, Google’s People Analytics Unit identified psychological safety as the number one characteristic of successful high-performing teams [40]. Other findings on psychological safety have shown the concept being influential in, e.g., job-commitment [41,42] and creativity (generation of novel ideas) [43].

#### 2.2.2. Relationship between Psychological Safety and Task Performance

Edmondson’s [9] influential article describing psychological safety’s impact on team learning and, consequentially, team outcomes, elicited a vast array of research pertaining to this domain, only fortifying the concept’s impact on team performance. Edmondson [9] found in her study of 51 decision-making teams that members learn from their mistakes to a greater extent when the levels of psychological safety are perceived to be high, and that team learning is positively related to team performance. In a study investigating the implementation of new technology in 16 cardiac surgery teams, Edmondson and colleagues found that the teams that developed psychological safety to be more collaborative [44].

Several researchers have contributed to this growing literature. Baer and Frese [13] found a positive relationship between psychologically safe environments and firm performance. Simons and Peterson [45] investigated intragroup trust (conceptually similar to psychological safety) in top-management teams and found that teams with a high level of trust were better to distinguish relationship conflicts from task conflicts—in contrast to those teams with lower intragroup trust. Kostopoulos and Bozionelos [46] found that psychological safety promoted two kinds of learning—exploratory and exploitative—that again were related to team performance. Bradley et al. [14] proposed how psychological safety could lead to better team performance, and their study found how psychological safety moderated the relationship between task conflict and performance. When psychological safety was perceived as high, task conflict and performance were positively associated. Additionally, Carmeli, Tishler and Edmondson [47] linked conditions of trust created within the team to improved decision quality and better performance in their study of CEO relational leadership and strategic decision quality in top management teams.

In Edmondson and Lei’s review, compiling six decades of research and history, they conclude that “in sum a psychologically safe environment enables divergent thinking, creativity, and risk-taking and motivates engagement in exploratory and exploitative learning, thereby promoting team performance” [10]. This was further supported in a systematic review on psychological safety, where Newman et al. [48] highlighted the positive outcomes for individuals, teams and organizations associated with psychological safety. This is consistent with Frazier and colleagues’ comprehensive meta-analytic review on the antecedents and outcomes of psychological safety, where psychological safety was found to be positively associated with task performance, as well as employee engagement, satisfaction and commitment [39].

Importantly though, Sanner and Bunderson [49] has furthered this understanding by contesting psychological safety being as effective regardless of the team’s task at hand. Considering that “the implicit assumption seems to be that teams are teams” [49], they point to the fact that the magnitude of the relationship between psychological safety and team performance varies across several studies. They propose a more nuanced view, where psychological safety becomes less important for tasks where problem-solving and information-sharing are less central to success. Sanner and Bunderson found that teams that work in “cognitively simple, isolated, and routine tasks where the ends and means of production is clear” [49] do not engage in experiential learning processes, and thus, do not benefit much from psychological safety and its mitigation of perceived interpersonal risk. Moreover, when teams rely more on social interaction, sharing perspectives and collective problem-solving, dubbed knowledge-intensive work, psychological safety is a more important and enabling condition [49].

In this study, our aim is to study management teams where members are interdependent and their responsibilities often require interpersonal activities, such as informational exchange, cross-functional collaboration and decision-making. Thus, from Sanner and Bundersons’ viewpoint on task-based contingency, psychological safety should, to a greater extent, impact team behaviors and performance in these teams.

#### 2.2.3. Relationship between Psychological Safety and Individual Satisfaction

The literature on individual satisfaction points to the concept being understood in a variety of ways, some pertaining primarily to the affective state (positive affects compared to negative affects), whereas other encompass a broader domain of behaviors (e.g., autonomy, personal mastery and positive interpersonal relations) and motivation [50,51]. Being employed has in itself been deemed as an important factor associated with individual satisfaction [52], and thus, the concept of satisfaction and how it is understood is something that has caught the interest of researchers, especially in work contexts [51]. We follow the compelling definitions of Hackman [28] and consider a more multidimensional approach, incorporating both the affective and non-affective aspects of satisfaction—namely, learning, well-being and growth. Clearly satisfaction relates to a wide range of concepts, and thus, holds the potential to be impacted by several factors concerning the interdependent processes and states emerging within a management team—one of them being psychological safety.

The aforementioned studies have already displayed psychological safety’s influence on team members’ learning, and how this consequentially could lead to better performing teams [9,10,44]. We suggest that team members’ belief that it is safe to engage in open informational exchange, speak one’s mind, voice concern and admit mistakes, will lead to individual well-being, learning and growth. Research on satisfaction has suggested that employees who perceive a lower risk in making mistakes are more likely to be satisfied with their work [53]. Similarly, when an employee feels the environment to be sufficiently safe, they are also more likely to commit to their work, and thus, psychological safety holds the potential to result in higher levels of job commitment [39,41,42]. Overall, psychological safety seemingly impacts both affective and non-affective states in individuals’ perceived satisfaction.

**H1**. 
*Psychological safety is positively related to task performance.*


**H2**. 
*Psychological safety is positively related to individual satisfaction.*


### 2.3. Behavioral Integration

#### 2.3.1. Past Research on Behavioral Integration

The concept of behavioral integration results from a long lineage of research trying to identify characteristics associated with highly functioning management teams. Previous research on variables predicting top management team performance, directed much of its attention towards demographic variables explaining outcomes [54]. The argument was that one could use demographic characteristics of team members as proxies for individual attributes (personality, attitudes, values) and explore the relationship between team member attributes and firm performance indicators. Critique was later raised, pointing to the approach having a number of limitations, and particularly what Lawrence [55] referred to as the “black box” problem. She argued that emphasizing prediction over explanation, thus, ignoring more subjective and process related concepts, risked leaving out numerous interpretations of the relationship between the demographic predictors and outcomes. Lawrence argued that this research had ultimately failed to consider these crucial intervening processes, further criticizing that subjective concepts in general had been left “unmeasured and untested” [55].

Whilst researchers sought to fill in the gap of the black box, trying to specify these intervening mechanisms by pointing to single dimensions like, e.g., communication frequency [56], social integration [56], collaboration [57] and interdependence [58], Hambrick [15] devised a more inclusive construct encompassing the complexities of management team interaction processes that could not be easily captured by any single-process dimension. Behavioral integration refers to “the degree to which the group engages in mutual and collective interaction” [15]. It incorporates one social and two task dimensions: (1) The level of collaborative behavior between members; (2) The quality and quantity (speed, accuracy and richness) of information exchange; (3) Joint decision-making [16].

Following this line of thought, the extent to which a team is fully integrated is gauged by those three components. This discerns it from what researchers deemed as mere work groups [59], where members simply acted in parallel on relatively discrete tasks. Evidently a management team is considered well integrated when the team members share vital information, engage in collective decision-making and collaborate closely with each other. Behavioral integration, thus, with its inclusion of several intervening processes focusing on a more multidimensional approach, renders a greater understanding of the dynamics within a management team than accounted for by distal demographic predictors or single process dimensions.

The multidimensional rationale of behavioral integration has been acknowledged by later research [60], where empirical findings have associated behavioral integration with, e.g., employee work outcomes [19], teams’ decision quality [18] and negatively to team affective conflicts [20].

#### 2.3.2. Antecedents for Behavioral Integration

What stimulates management teams to become behaviorally integrated? What makes them more susceptible to engage in open information exchange and collaboration-based decision-making? Simsek et al. [60] tried to work out some of the antecedents shaping behaviorally integrated teams by conducting a multilevel analysis, using data from 402 firms. Their findings include several multilevel determinants, such as CEO tenure, size of management team, team level educational diversity and firm performance [60]. Other researchers focused primarily on CEO influence, leaders’ expectations and supportive behaviors to facilitate integration [61,62], whilst Carmeli and Shteigman [63] applied a social identity perspective. They build on the concept of social identification processes that shifts the perspective of it as an individual phenomenon, and argue for collective team identification [64], treating it as a group-level phenomenon. Collective team identification refers to members’ shared sense of identification with a work group, and the “emotional significance that members of a given group attach to their membership in that group” [64]. Thus, they acknowledged the cognitive, evaluative and emotional aspects of identification, in line with traditional social identity theorists. It is a shared sense of identification held by those in the management team.

How is this meant to facilitate collaborative interactions and informational exchange in management teams? Carmeli and Shteigman [63] leaned on Turner in that social identity is the cognitive mechanism that facilitates group behavior, and proposed that when teams develop strong identification, “they redefine the self as ‘we’ rather than ‘I’ and share common ground” [65]. They posit that “members are likely to value and work for a collective purpose rather than their individual interests when a strong collective identification emerges” [65]. This can make members more inclined to engage in collaborative behavior, joint decision-making and informational exchange. Their findings support this, showing that collective team identification indeed facilitates behavioral integration. By providing insights into the underlying mechanisms in a management teams’ dynamics through social identity theory, their findings also entertain and support the notion that identification forms a motivation to engage in a given task [66].

#### 2.3.3. Relationship between Behavioral Integration and Team Effectiveness

Indeed, there has been some empirical findings of outcomes linked to behavioral integration. Carmeli [17] found that top management teams engaging in mutual and collaborative interaction do have a positive impact on firm performance levels. In addition, Carmeli and Halevi [67] showed how processes of behavioral integration within the management team shape more competent organizations, and Carmeli and Schaubroeck [18] found behavioral integration to be negatively associated with organizational decline.

In the same study, Carmeli and Schaubroeck also found that behaviorally integrated management teams were perceived to reach better quality strategic decisions than those perceived to be less behaviorally integrated. Mooney and Sonnenfeld [20] investigated behavioral integration’s association with team affective conflicts, considered to be more dysfunctional in the decision context of the management team, and found them to be negatively correlated with each other. Hambrick [68] also found that deficient performance was a result of poor behavioral integration leading to ineffective group decision processes in a medical product company led by a top management team. Additionally, Marks et al. [69] concluded their findings with interactions reflective of behavioral integration being strongly related to decision-making processes (performance), especially when the team embarked upon novel situations with high task difficulty. Overall, this is indicative of behavioral integration’s association with teams’ task performance (e.g., task results, decision quality).

Though there are some findings regarding behavioral integration’s impact on employee outcome in, e.g., how management teams’ influence others outside the team, leading to employee individual satisfaction [19], there is rather scarce research concerning these individual effects on the members within. Some research shows that joint decision-making, a task-related construct of behavioral integration, can increase motivation, job satisfaction and commitment [70,71]. We, therefore, posit that there is a positive association between the level of behaviorally integration in a team and the team members’ level of individual satisfaction. 

**H3**. *Behavioral integration is positively associated with task performance*.

**H4**. *Behavioral integration is positively associated with individual satisfaction*.

#### 2.3.4. Psychological Safety as a Possible Antecedent for Behavioral Integration

Compiling Edmondson’s research on psychological safety [10] combined with Simon and Peterson’s [45] findings on trust and its pivotal role in group processes can be indicative of an association between psychological safety and behavioral integration. Furthermore, psychological safety is documented to create a context in which it is easier to collaborate [36]. A central theme of this line of research is how psychological safety facilitates team members’ willingness to contribute with ideas and cooperative behaviors, leading to behaviors such as sharing information and knowledge between employees [10,72,73]. Those team members perceiving high psychological safety are more inclined to engage in such interpersonal risk behaviors; thus, focusing less on self-protection and more on the team as a whole. In sum, psychological safety is likely to affect the way members interact with each other [9].

The resulting collaborative tendency of psychologically safe environments may, thus, account for some of the integral parts in behavioral integration (collaborative behavior, joint decision-making and information exchange). Empirical research on both individual and group levels have found psychological safety to be positively related to the sharing of information [73,74], providing an environment in which collaboration and feedback seeking are accepted and encouraged [36]. Thus, we posit there will be a positive relationship between psychological safety and behavioral integration.

**H5**. *Psychological safety is positively associated with behavioral integration*.

#### 2.3.5. Behavioral Integration as a Mediator between Psychological Safety and Team Effectiveness

Carmeli points to the fact that neither does behavioral integration alone encompass “basic elements that make for positive interpersonal relationships such as respectful engagement, trust and task enabling, nor does it address the emotional space in which people flourish (…)” [17]. We suggest that psychological safety may represent such conditions where members feel comfortable in expressing themselves, and thus, make way for information-sharing and collaborative behavior to occur. One could also surmise from these safe environments that they would aid people to engage in more joint decision-making, and thus, strengthen a potential association between the two.

Psychological safety and behavioral integration have both been associated with team effectiveness. In addition, some studies have also looked at other factors potentially enabled by psychological safety and how they mediate its impact on team effectiveness. For instance, Edmondson’s study from 1999 [9] indicated that psychologically safe environments enabled team learning, which again contributed to team effectiveness. Hambrick [15] postulated that management teams are subject to centrifugal forces driving members apart, such as competing goals and interests, and that these forces hamper the team from becoming a collaborative and coordinated unit. If a management team was to operate as an integrated whole, it must support forces driving members together to behave as a real team. It is our hypothesis that psychological safety could act as one of those forces.

**H6**. *Behavioral integration mediates the relationship between psychological safety and task performance*.

**H7**. 
*Behavioral integration mediates the relationship between psychological safety and individual satisfaction.*


## 3. Method

### 3.1. Procedure, Design, and Sample

This study uses self-report data collected through a questionnaire called *effect.* The questionnaire is based on research described in the book *Effective Management Teams and Organizational Behavior: A Research-based Model for Team Development* [75], measuring several aspects of management team effectiveness and functioning. The management teams responding to the *effect* questionnaire were either a part of a development program designed for management teams or part of an assessment of how such teams in various Norwegian companies operate.

The sample of management teams used in this study comprised a total of 1150 leaders pertaining to 160 management teams from Norway, with a gender distribution of 50.1% being male respondents and 49,9% female. The data were collected from first-time respondents answering the questionnaire from March 2017 until October 2019. The team tenure distribution of the respondents was categorized as follows: less than one year (23.6%), 1–2 years (27.4%), 3–4 years (18.7%) and longer than 5 years (30.3%).

The teams participating were classified according to their organizational levels: top management teams (50.4%), middle management teams (31.9%) and lower-level management teams (17.7%). The teams were either from governmental or municipal agencies (43%) or from private or public enterprises (57%). Finally, team size varied from 3 to 19 members. The majority of the teams consisted of 5 to 8 members, and the average team size was seven members. 

### 3.2. Measures

The four variables measured in this study are psychological safety, behavioral integration and the two performance measures: task performance and individual satisfaction (see Appendix A for the items measuring the respective variables). Each variable consisted of 4 to 7 items, where the responding participants rated their management team on a 7-point Likert scale, from 1 (*strongly disagree)* to 7 *(strongly agree*). In addition, items could be scored as “do not know/not applicable.”

The individual scores on measures included in this study were aggregated from individual to team level data.

Reliability was estimated for all scales. Cronbach alpha is a general measure for the internal reliability of multiple items, assessing the consistency for the entire scale. An acceptable threshold for internal consistency is commonly set at 0.70 [76,77]. All Cronbach alpha-values for the scales in this study are estimated to be 0.70 or higher (α ≥ 0.70).

### 3.3. Variables of the Study

**Task performance.** Task performance was operationalized based on the work of Bang and Midelfart and measures the extent to which “the results produced by the team make a significant and positive contribution to the success of the organization–in other words that the results create substantial added value for the organization” [8] (p. 336). Task performance constitutes the direction, alignment and commitment a management team creates, as well as the high-quality decisions it makes. This is measured through the participants’ response to seven items (see Appendix A). Scale reliability was estimated at α = 0.92.

**Individual satisfaction.** The individual satisfaction measure is defined as the extent to which a “team contributes to the individual member’s well-being, motivation, learning and growth” [8]. This scale is based on Hackman’s [28] and Wageman et al. ‘s [7] conceptualization of individual well-being and growth. The scale consisted of five items (see Appendix A). Scale reliability was estimated at α = 0.86.

**Behavioral integration.** The scale measuring behavioral integration builds on the work of Hambrick [15,16] and Carmeli and colleagues [17,18,60]. Behavioral integration is defined as the degree to which the management team engages in “mutual and collective interactive processes through which it displays information exchange, collaborative behavior, and joint decision-making” [17]. The scale consists of five items (see Appendix A). Scale reliability was estimated at α = 0.82.

**Psychological safety.** The psychological safety measure was based on the work of Amy Edmondson [9,10,11], measuring if it is perceived as safe to speak up, express uncertainty and ask for help in the management team. This measure originally consisted of five items, but due to cross-loadings with behavioral integration one of the five items was deleted (see Table 1 and Table 2). The four items’ reliability was estimated at α = 0.90.

To ensure that behavioral integration and psychological safety were distinct and represented separate concepts, a principal component analysis (PCA) was conducted on the 9 items measuring these two variables, using oblique rotation (direct oblimin). The groups of items retained for the PCA are indicated in Table 1. One item in the rotated factor matrix cross-loaded on more than one factor. Item 3: *“It is easy to ask other management team members for help”*, which loaded at −0.37 on psychological safety, also loaded on behavioral integration at 0.39. Tabachnick and colleagues [78] suggest that a minimum loading of an item at 0.32 is as good rule of thumb when considering if an item is to be dropped from further analysis. This equates to approximately 10% overlapping variance with the other items in that factor [79].

Furthermore, it is suggested to employ an alternative rotation method to see if the cross-loading persists in another rotation method, and thus, define a simpler structure [79]. An orthogonal rotation produced results nearly identical to the oblique rotation when using the same extraction method. There is a cross-loading that loads higher than 0.32 on the two variables of psychological safety and behavioral integration that persist through different rotations. Hair et al. [80] suggest that when such cross-loadings persist, as is the case for this item, they become candidates for deletion. Thus, we removed Item 3 from the analysis.

### 3.4. Aggregation

This study’s aim is to infer different group phenomena in management teams. Our data are obtained through the collection of individual responses to a questionnaire; thus, aggregation of data is necessary. The questionnaire is designed in such a way that each item is framed to represent questions pertaining to the team level phenomenon. Respondents were asked to rate statements worded as if by observers of the team, e.g., Item four in the team performance scale: *“The management team ensures that goals and processes are coordinated and consistent”,* and Item two on the scale measuring behavioral integration: “*The members of the management group have a clear understanding of each other’s issues and needs*”. This ensures the observations perceived by the individual to be situated at a team level. The process of aggregation offers the opportunity for the individual team member’s response to be merged with other members and analyzed at a team level. Accordingly, whether the rater observations are similar enough and applicable to represent a homogenous whole [81], thus, reflecting a group phenomenon, is a legitimate issue at hand. To justify whether it is meaningful to initiate this process converting individual responses by aggregation, yielding team level data, we conducted analysis of interrater agreement (*rWG*) and interrater reliability (intraclass correlation coefficient (ICC) [82,83]. 

#### 3.4.1. Interrater Agreement

The average within-group rater agreement (*rWG*) is a measure of agreement between group members and, in this study, pertaining to their response to each item compromising the variable in question. *rWG* quantifies the extent to which multiple judges’ ratings are interchangeable due to their absolute similarity [82]. It assesses this agreement by comparing the observed variance from multiple raters “to the variance expected when there is complete lack of agreement among raters (i.e., random responding)” [82]. It is, thus, central to obtain justifiable within-group-agreement, through the *rWG*, for us to infer the group mean as reflective of a group phenomenon.

There has been a debate on what ought to constitute the acceptable level of agreement, and thus, a tolerable threshold level for the *rWG* [84]. One favorable view is to consider the threshold level on a continuum agreement scale ranging from 0.00–1.00, where 0.00–0.30 equals “lack of agreement”, 0.31–0.50 = “weak agreement”, 0.51–0.70 = “moderate agreement, 0.71–0.90 = “strong agreement” and 0.91–1.00 = “very strong agreement” [82,83]. Biemann et al. [82] argue that one should consider the type of research at hand before analyzing and setting the final cut-off points of the *rWG*. Due to the nature of the phenomena being studied, it is reasonable to expect that some variance between the different respondents’ perceptions will occur, when assessing the team processes and outcome variables. In studies researching general trends across multiple teams, it has been argued that it would be sufficient to obtain a moderate agreement between individual respondents, reflecting a cut-off point at 0.50 = “moderate agreement” [82].

Although excluding teams exhibiting a low level of within-agreement seems like a preferable option, Biemann et al. [82] advise against this. They argue that the study risks losing statistical power, and that the variance observed may in fact be valuable and originate from natural dissimilarities rather than measurement error. Several approaches have been proposed to mitigate the risk of groups with low-level within-agreement imposing a threat to the analyses conducted. LeBreton and Senter [83] offer one approach, encouraging researchers to calculate the respective *rWG*s for all groups included in the study. An examination of the percentage of groups exceeding a cut-off of 0.70 should be conducted, and if a high percentage is found, one should include the low-level groups. Conducting such an analysis using this approach, the interrater agreement mean (mean *rWG*) should be above cut-off point, which, in this study, is set at 0.50 (moderate agreement). Table 3 shows a *rWG* mean for our groups exceeding the cut-off point at 0.50 for all the included variables, with a high percentage of the *rWG* group values also surpassing this threshold (ranging from 76,9% to 86,09%).

#### 3.4.2. Interrater Reliability (ICC)

Another analysis conducted to justify aggregation of our data, estimating the reliability between team member scores and management team scores, is the intraclass coefficient correlation ICC(2). It takes into account the variance occurring within each group and between each group responding to the variable in question, and is often interpreted “as the proportion of observed variance in ratings that is due to systematic between-targets differences compared to the total variance in ratings” [83] (p. 822). By examining this proportion of scores attributed to between-team variance compared to what is attributed to variance within-team scores, it provides a valuable measure indicating how meaningful it would be to aggregate scores from individual level to team level. 

Furthermore, generating a sensible cut-off score for the ICC(2) has been a topic of discussion. As LeBreton et al. [85] found in their research, reported values of ICC(2) tend to be lower in the domain pertaining to organizational research. They suggest that this is due to restrictions within measures of certain variables rather than individuals differing on their response to the same phenomenon. Akin to the *rWG*, Biemann et al. [82] encourage setting a cut-off point suitable for the specific nature of the research domain and study context. Moreover, in their guidelines for selecting intraclass coefficients for reliability research, Koo and Li argue for a general guideline of “values between 0.5 and 0.75 indicate moderate reliability and values between 0.75 indicate good reliability and values greater than 0.90 indicate excellent reliability” [86] (p. 161). A high ICC(2) level represents a high between-team variance and a low within-team variance. We apply a cut-off point at 0.5, setting a moderate level of agreement for all our measures for the ICC(2) values. The ICC(2) values in this study range from 0.68 to 0.73 (see Table 3).

#### 3.4.3. Data Analysis

All statistical analyses in this study were performed in SPSS (27th ed.). We conducted correlation analysis for *H1*, *H2, H3, H4* and *H5*. Linear regression analysis with the macro PROCESS written by Andrew F. Hayes [87] was used for our model estimation of *H6* and *H7*. PROCESS provides the regression coefficients in a simple mediation model, and bootstrap confidence interval for inference of the indirect effects’ statistical significance [87]. Bootstrapping is a statistical method of simulation that aims to create a sample distribution by repeatedly resampling a single data set, in this case the indirect effects, as a means to mimic the original sampling process [87]. The indirect effect of X on Y through M is the product of *path a* multiplied with *path b* (see Figure 1). A bootstrap confidence interval for the indirect effect (ab) is generated after “randomly resampling *n*-cases from the data with replacement, where *n* is the original sample size in the study, and estimating the model and resulting indirect effect *ab* in this bootstrap sample” [88]. A bootstrap confidence interval of 95% has two endpoints, an upper limit (ULCI) and a lower limit (LLCI). The true value of an indirect effect resides with 95% certainty between these endpoints in our bootstrap sample [76].

An indirect effect is statistically significant when the values between the upper (ULCI) and lower limits (LLCI) of the bootstrap interval do not straddle zero [88]. The procedure of resampling is often repeated thousands of times. In this study we ran 5000 bootstraps. This allows us to generate a sound empirical representation of the sampling distribution of the indirect effect. This computation is argued to be one of the more valid and powerful methods for testing intervening variable effects in simulation research [89,90], and is considered adequate to support a claim of an indirect effect of X on Y, mediated through M, if the indirect effect proves statistically significant [87]. Figure 1 is a visual depiction of a simple mediation model. The relationship between *X* and *Y* is denoted by *c* (total effect), whereas *c’* (direct effect) concerns the relationship between these variables when controlled for by the mediator (M). Another way of calculating the indirect effect is to subtract the *c*’ from the total effect *c* [87].

#### 3.4.4. Control Variables

Becker and colleagues [91] argue that there must be a clear and defensible purpose when including control variables in further analysis, with several researchers stating that the basis for inclusion should lie in theory [92,93,94]. When lacking such a convincing rationale of justification, this should result in the exclusion of these variables [91]. 

**Management team level (MTL).** Data were collected from management teams at different levels in the organizational hierarchy. Floyd and Lane [95] propose that management teams pertaining to different levels (top, middle and lower level) are inherently different in their ways of processing information. Depending on level allocation, it is argued that management teams process unique types of information and engage in different forms of behaviors, and thus, there could be a distinctive form of motivation based on level allocation.

Top management teams, due to their vital role in the organization, might be more motivated to set direction, make good decisions and be effective [95,96]. Middle management teams on the other hand, are argued to have a role of conveying information from the top to the operating level (lower-level management teams), whilst the latter primarily reacts to information, e.g., from top level and feedback from the market their products addresses [95]. Additionally, top management teams have the tendency to consist of more competitive, individualistic and ambitious managers [15,97], potentially having less time tending to interpersonal relationships and pressing members to be highly effective.

Thus, considering these combined differences in characteristics and the information being processed within each management team, it is reasonable to assume that this could impact team dynamics and performance. Accordingly, MTL could potentially influence both psychological safety and behavioral integration, in addition to the performance measures included in our study. Level was measured using an ordinal scale with items ranging from 1–3, with 1 referring to the top management team, 2 to the team directly below the top management team and level 3 being a team ranking lower than the former in the organizational hierarchy. Hence, MTL is suitable for further analysis and subsequently considered if it ought to be included as a control variable.

**Team size.** Previous research has argued for a team potentially experiencing reduced productivity based on the number of members it holds [98]. Steiner [99] as cited in Forsyth [98] referred to this phenomenon as process loss, meaning a reduction in the performance due to “action, operations or dynamics” that prevent the group from reaching its full potential [98] (p. 320). This follows Max Ringelmann’s [100] earlier findings of the tendency of people to become less productive when they work with others, and that this loss of productivity increases as the size of the group increases. Team size itself could also potentially be a source of confrontations, large group effects and logistical issues, such as finding enough physical space and time to meet [101]. Depending on factors such as a team’s purpose and type of tasks, optimal team size will vary. Wageman et al. [7] suggested that the best management team typically consisted of no more than eight members.

Thus, team size could affect both the outcome variables (task performance and individual satisfaction), in addition to psychological safety and behavioral integration. In the present sample, our management teams ranged in size from 3 to 19 members. Thus, we will include team size in our further analysis, and subsequently, consider its inclusion as a potential control variable.

## 4. Results

### Main Analysis

Means, standard deviations and bivariate correlations are presented in Table 4. To examine the potential impact of the control variables management team level and team size, a correlation analysis was performed including all variables in the study (see Table 4). Management team level showed a significant association with the two outcome variables (*r* = −0.16, *p* > 0.05 for task performance and *r* = −0.17, *p* > 0.05 for individual satisfaction). We also found team size to be correlated with the mediator behavioral integration (*r* = −0.20, *p* > 0.05). Hence, we included management team level and team size as control variables in the subsequent regression and bootstrap analysis.

Hypotheses 1 and 2 (see Figure 2) predicted a positive relationship between psychological safety and the two outcome variables. Analyses showed a significant positive bivariate correlation between psychological safety and task performance (*r* = 0.51, *p* ≤ 0.01), and between psychological safety and individual satisfaction (*r* = 0.70, *p* ≤ 0.01). When controlling for team size and MTL, as shown in Table 5, the partial correlations were almost similar to the bivariate ones, and psychological safety was still positively correlated with task performance (*r* = 0.51, *p* ≤ 0.01) and with individual satisfaction (*r* = 0.72, *p* ≤ 0.01). Hypotheses 1 and 2 were, therefore, supported.

Hypotheses 3 and 4 predicted that there is a positive relationship between behavioral integration and the two outcome variables. We found a positive bivariate correlation between behavioral integration and task performance (*r* = 0.77, *p* ≤ 0.01), and between behavioral integration and individual satisfaction (*r* = 0.76, *p* ≤ 0.01). When controlling for team size and MTL, as shown in Table 5, the partial correlations were marginally lower than the bivariate ones, and behavioral integration was still positively correlated with task performance (*r* = 0.76, *p* ≤ 0.01) and with individual satisfaction (*r* = 0.75, *p* ≤ 0.01). Hypotheses 3 and 4 were, therefore, supported.

Hypothesis 5 predicted a positive relationship between psychological safety and behavioral integration. Analyses showed that there was a significant positive bivariate correlation between psychological safety and behavioral integration (*r* = 0.68, *p* ≤ 0.01). When controlling for MTL and team size, as shown in Table 5, the partial correlations were marginally lower than the bivariate ones, and psychological safety was still positively correlated with behavioral integration (*r* = 0.67, *p* ≤ 0.01). Hypothesis 5 was, therefore, supported.

**Mediation analysis.** Hypotheses 6 and 7 predicted that behavioral integration mediates the relationship between psychological safety and the two outcome variables: task performance and individual satisfaction. In a model with behavioral integration as the mediator, we found a significant indirect effect (IE) of psychological safety on task performance through behavioral integration (*IE* = 0.51, 95%Cl (0.3683–0.6872)). There was also a significant indirect effect of psychological safety on individual satisfaction through behavioral integration (*IE* = 0.33, 95%Cl (0.2270–0.4526)) (see Table 6). When controlling for team size and MTL, as shown in Table 7, the indirect effect of psychological safety on task performance through behavioral integration was marginally lower but significant (*IE* = 0.50, 95%Cl (0.3573–0.6684)). When controlling for team size and MTL, the indirect effect of psychological safety on individual satisfaction through behavioral integration was marginally lower but significant (*IE* = 0.30, 95%Cl (0.2014–0.4220)). Hypotheses 6 and 7 were, therefore, supported.

We found no direct effect of psychological safety on task performance (*B* = −0.01, ns), but a positive significant direct effect (*B* = 0.32, *p* ≤ 0.01, 95%Cl (0.2016–0.4368)) of psychological safety on individual satisfaction. When controlling for team size and MTL, there was still no direct effect of psychological safety on task performance (*B* = −0.01, ns), while the direct effect of psychological safety on individual satisfaction was marginally higher (*B* = 0.35, *p* ≤ 0.01, 95%Cl (0.2016–0.4368]). Note that *B*, as provided by the bootstrap analysis, represents the unstandardized regression coefficient.

## 5. Discussion

This study explores the relationship between psychological safety, behavioral integration, and team effectiveness in management teams. Prior research has recognized that both psychological safety and behavioral integration are positively related to team performance in general, and our findings indicate that this is also true for management teams. In addition, we found that behavioral integration mediates the relationship between psychological safety and two outcomes—task performance and individual satisfaction—in management teams.

### 5.1. Theoretical Implications

Our study seeks to contribute to the existing literature in several ways. First, our findings extend the prevailing framework of psychological safety, strengthening its empirical support by showing that it plays an important part in team outcomes in management teams. The field of research on psychological safety has previously been occupied by studies conducted in healthcare units and work team settings [9,10,11,102,103], and very few studies have been conducted on management teams. Thus, our findings add to the existing literature by expanding the applicability of psychological safety and its influence on performance, with positive outcomes also yielded in management teams. Management teams are important in setting goals and direction, as well as offering strategies deemed important for the organization and its overall performance. 

Second, our findings contribute to the literature of psychological safety by unfolding its little-known relationship with behavioral integration. As exhibited in our theoretical framework, psychological safety is documented to create a context in which it is easier to collaborate [36]. It facilitates the willing contribution of ideas and actions, ultimately leading to behaviors such as the sharing of information and knowledge between employees [10,72,73]. Our study is the first to point to a positive relationship between behavioral integration and psychological safety in management teams. Thus, the more team members perceive the climate as safe in terms of speaking their mind without fear of repercussions, the more they partake in mutual collaboration, information sharing and experience ownership in the decisions being made.

Furthermore, our findings suggest that the behaviors characterizing behavioral integration may explain how psychological safety influences task performance and individual satisfaction in management teams. Psychological safety is likely to impact the way team members interact with each other [9], being more inclined to engage in interpersonal risk-taking, when feeling psychologically safe to do so. Thus, focusing less on self-protection and more on the team as a whole, seemingly affecting and enabling the ability of teams becoming more integrated and accomplish team effectiveness. Hambrick [15] postulated that management teams are subject to forces driving members apart, such as competing goals and interests, and that these forces hamper the team from becoming a collaborative and coordinated unit. In line with Hambrick, we argue that for a team to become an integrated whole it must support forces steering them towards behaving as a real team. Based on our findings, we believe psychological safety to be one of those forces.

Third, though there are some findings of behavioral integration impacting employees’ individual satisfaction outside a management team [19], there is scant research concerning these individual effects on the members within. Our results point to behaviorally integrated teams not only being associated with performing better on task-related outcomes, but also individuals reporting more satisfaction being part of that management team.

Lastly, the concept of behavioral integration originates from the study of top management teams [15]. These differ from other management teams in their constellation of the most influential executives situated at the apex of an organization. Our sample consists of management teams at different levels, and thus, expands the applicability of behavioral integration’s impact on team performance to management teams at all levels in the organizational hierarchy.

### 5.2. Implications for Practice

Our research may provide several implications for practice. The findings show that behavioral integration explains some of the impact psychological safety has on performance in management teams, and thus, highlights the influential role of psychological safety and its relationship with behavioral integration in creating conditions under which effective management teams seem to thrive. Hambrick [15] argued that a well-designed and functioning management team is one that is behaviorally integrated, in the sense that it works as a team. Thus, conversely, a management team exhibiting low levels of psychological safety, associated with lower levels of behavioral integration, will not function particularly well as a team.

Our findings should, therefore, encourage management teams at all levels and their members towards becoming more conscious of promoting psychological safety, bearing in mind that this could potentially influence the effectiveness and how integrated a team is. Considering management teams’ central position in an organization, setting directions, offering strategies and engaging in complex decision-making, it is reasonable to think that functioning as a team (behavioral integration) would be key to accomplishing these tasks. Additionally, team members feeling safe to speak up, voicing their concerns and not withholding potentially unique and important information, would aid a management team to perceive themselves as one. Our results should encourage management teams at all levels to nurture those aspects of team engagement and interactions that lead to psychologically safe environments.

### 5.3. Limitations

There are multiple methodological limitations to be considered. First, being a cross-sectional study, with data collected at one specific given point in time, and with the same team members evaluating both the predictor and criteria variables, makes it hard to claim causality. We describe the association between psychological safety, behavioral integration and team performance as a causal relationship, but this type of study cannot demonstrate cause and effect [87]. Our data show correlations between variables in this study, but the correlation coefficients give no indication of the direction of causality. We cannot say whether high levels of psychological safety led to high levels of behavioral integration or whether high levels of behavioral integration led to high levels of psychological safety. There is an underlying methodological bias contingent on the non-established temporal sequence in these studies, which implies that the associations being reported should be cautiously considered.

Second, being based solely on quantitative methods, this provides another potential methodical challenge for our study. A systematic review of the literature of psychological safety [48] points out the need for alternative methodologies to study psychological safety. The methodology in existing research is a skewed towards a quantitative survey methodology, and by considering other alternatives one can gain “a more holistic understanding as to how psychological safety develops and influences work outcomes.” [48]. Longitudinal data collection could provide a stronger theoretical foundation than cross-sectional data. One opportunity is longitudinal research that allows both a better assessment of cause and effect and an examination of changes in psychological safety over time [10].

Third, the variables used in our study are based on the self-reported measures. This causes a potential methodological challenge because the respondent might be motivated to respond in a socially desirable manner [104]. Social desirability bias is the tendency of respondents to answer in a way deemed more socially acceptable, doing this to “project a more favorable image of themselves and to avoid negative evaluations” [105]. We must be aware of the potential social desirability tendency since all variables in this study were self-report and part of the same questionnaire.

Lastly, Edmondson and Lei [10] suggest performing cross-cultural comparisons of relationships between psychological safety and performance outcomes. They point out that in some cultures it could be difficult to ask questions, have a discussion with team members and provide feedback to members of the team. They argue that this could be interpreted as rude and inappropriate behavior. The cultural differences mentioned show the importance of studying the association in different cultures. Our study consists of a relatively large sample of 1150 respondents, but they were not randomly assigned and include entirely Norwegian management teams. This opts for future research to replicate this study in other cultures.

### 5.4. Recommendations for Future Research

Future research should take these limitations into consideration. First, to better understand the association between psychological safety, behavioral integration and team performance, longitudinal data collection could contribute to a stronger theoretical foundation than cross-sectional data. As pointed out by Edmondson and Lei [10], longitudinal research can examine the changes in psychological safety over time and allow a better assessment of cause and effect. Due to the skewness towards quantitative research conducted on the phenomenon of psychological safety, they argue using hybrid methods, mixing qualitative and quantitative data in future studies.

Second, Edmondson and Lei [10] point out that one of the further research directions and theoretical opportunities when it comes to psychological safety, is to perform cross-cultural comparisons of relationships between psychological safety and performance outcomes. As discussed in the limitations, all management teams in this study were based in Norway. Thus, we echo these findings where we suggest that our study should be applied in a cross-cultural manner. Additional research is needed to test if our findings could be replicated in and generalized across other cultures.

Third, in our study we have investigated a relationship that has added empirical data to the existing research on psychological safety. We will draw attention to the importance of replicating and further investigating these variables as valid predictors to be conducted in future team studies. Future research should also keep exploring an even broader range of mediating mechanisms through which psychological safety impacts a management teams’ performance.

## 6. Conclusions

This study explores whether there is an indirect effect of psychological safety on team effectiveness in management teams, operating through the mediating variable of behavioral integration. Our findings support such an indirect effect. We are the first to point to a mediating effect of behavioral integration in the psychological safety and team effectiveness relationship in management teams. The more team members perceive their climate as safe in terms of speaking their mind without the fear of repercussions, the more they partake in mutual collaboration, information-sharing and experience ownership in the decisions being made. This is associated with management teams performing better – both when it comes to adding value to the enterprise and to contribute to individual well-being and growth for the team members. Our results suggest that psychological safety is a potential enabler of environments where behaviorally integrated teams could thrive. Additionally, the results point to behavioral integration as a possible explanatory mechanism for some of the observed difference in performance between management teams exhibiting high psychological safety and those who do not. This study adds to the existing literature by expanding the applicability of psychological safety and its influence on performance to management teams at all levels in the organizational hierarchy. Based on our findings, we encourage management teams and their members to have a keen focus on fostering psychologically safe environments, bearing in mind how this could impact how integrated a management team is, and how well it will perform.

## Figures and Tables

**Figure 1 ijerph-20-00406-f001:**
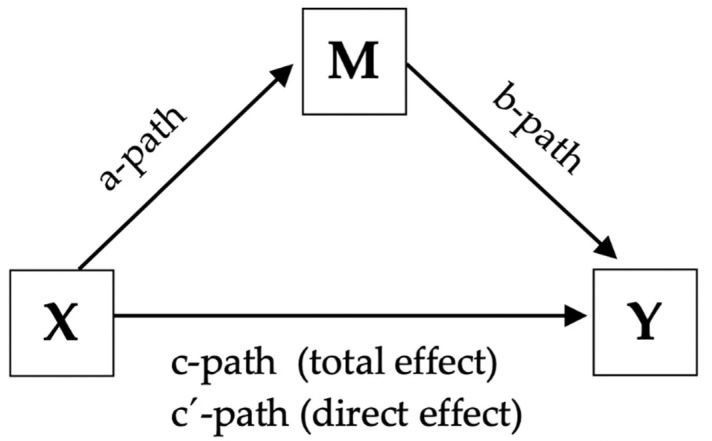
A conceptual diagram of a simple mediation model.

**Figure 2 ijerph-20-00406-f002:**
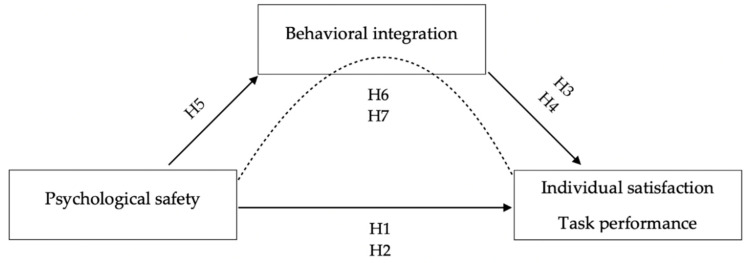
A conceptual model of our hypotheses and the relationship between the variables in this study.

**Table 1 ijerph-20-00406-t001:** Results from a principal component analysis of the two scales measuring Behavioral integration and Psychological safety.

Items	Factor Loadings	
	1	2
Factor 1: Behavioral integration		
Item 3	0.83	
Item 4	0.78	
Item 2	0.68	
Item 5	0.61	
Item 1	0.49	
Factor 2: Psychological safety		
Item 3	0.39	−0.37
Item 4		−0.92
Item 2		−0.86
Item 5		−0.77
Item 1		−0.73

Note. *n* = 1150. The extraction method was a principal component analysis with an oblique rotation (Direct Oblimin with Kaiser Normalization). All items can be found in Appendix A.

**Table 2 ijerph-20-00406-t002:** Results from a principal component analysis of the two scales after removing Item 3 from the Psychological safety scale.

Items	Factor Loadings	
	1	2
Factor 1: Behavioral integration		
Item 3	0.82	
Item 4	0.78	
Item 2	0.74	
Item 5	0.57	
Item 1	0.54	
Factor 2: Psychological safety		
Item 4		−0.92
Item 2		−0.87
Item 5		−0.76
Item 1		−0.74

Note. *n* = 1150. The extraction method was principal component analysis with an oblique rotation (Direct Oblimin with Kaiser Normalization).

**Table 3 ijerph-20-00406-t003:** Cronbach’s alpha, mean rWG, rWG percentage, ICC(2) for all scales.

Variable	Alpha	*rWG*	*rWG*% ^a^	ICC(2)
Psychological safety	0.9	0.68	76.9	0.67
Behavioral integration	0.82	0.73	83.1	0.68
Individual satisfaction	0.86	0.75	86.9	0.71
Task performance	0.92	0.68	76.9	0.73

^a^ Percentage of teams with an rWG-value > 0.50.

**Table 4 ijerph-20-00406-t004:** Means, standard deviation and bivariate correlations for all variables. *n* = 160.

Variable	Mean	SD	1	2	3	4	5	6
1. PS^.a^	5.44	0.68	−					
2. BI^.b^	5.1	0.63	0.68 **	−				
3. IS^.c^	5.23	0.63	0.70 **	0.76 **	−			
4. TP^.d^	4.88	0.66	0.51 **	0.77 **	0.80 **	−		
5. MTL^.e^	1.74	0.79	0.05	−0.12	−0.17 *	−0.16 *	−	
6. TS ^f^	7.19	2.71	−0.11	−0.20*	−0.12	−0.04	−0.22 **	−

^a^ Psychological safety ^b^ Behavioral integration. ^c^ Individual satisfaction ^d^ Task performance. ^e^ Management team level ^f^ Team size. * *p* ≤ 0.05. ** *p* ≤ 0.01.

**Table 5 ijerph-20-00406-t005:** Means, standard deviation and partial correlations for variables in H1, H2, H3, H4 and H5 controlled for team size and management team level. *n* = 160.

Variable	Mean	SD	1	2	3	4
1. PS^.a^	5.44	0.68	−			
2. BI^.b^	5.1	0.63	0.67 **	−		
3. IS^.c^	5.23	0.63	0.72 **	0.75 **	−	
4. TP^.d^	4.88	0.66	0.51 **	0.76 **	0.79 **	−

^a^ Psychological safety ^b^ Behavioral integration. ^c^ Individual satisfaction ^d^ Task performance. ** *p* ≤ 0.01.

**Table 6 ijerph-20-00406-t006:** Indirect effects and bootstrap results.

	Estimate	SE	95% CI	
			LL	UL
Indirect effect of psychological safety on task performance	0.51	0.08	0.37	0.69
Indirect effect of psychological safety on individual satisfaction	0.33	0.06	0.23	0.45

Note. Standard errors and confidence intervals estimated by 5000 bootstrap replications. CI = confidence interval; LL = lower limit; UL = upper limit.

**Table 7 ijerph-20-00406-t007:** Indirect effects and bootstrap results controlled for team size and management team level.

	Estimate	SE	95% CI	
			LL	UL
Indirect effect of psychological safety on task performance	0.5	0.08	0.36	0.67
Indirect effect of psychological safety on individual satisfaction	0.3	0.06	0.2	0.42

Note. Standard errors and confidence intervals estimated by 5000 bootstrap replications. CI = confidence interval; LL = lower limit; UL = upper limit.

## Data Availability

The data presented in this study are available on request from the corresponding author.

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
