# Peer review of "The Relationship between Psychological Safety and Management Team Effectiveness: The Mediating Role of Behavioral Integration"

_ijerph, 2022, doi:10.3390/ijerph20010406_

Round 1

Reviewer 1 Report

The paper is well written in detail. The scientific question posed in the paper is quite excellent and interesting. The paper reviews the previous literature quite well and the original contribution of the paper is well clarified. The analytical method is appropriate, and the results is meaningful. 

I found the paper is publishable at this version.

The research investigates the question whether there is an indirect effect of psychological safety on team 9 effectiveness in management teams. Especially, this paper examined the potential mechanisms that can explain this association in management teams by analyzing the role of behavioral integration for generating the relationship between team psychological safety and management team effectiveness.

This paper found that behavioral integration mediates the relationship between psychological safety and two outcomes – task performance and individual satisfaction – in management teams. Technically, the paper analyzes the regression model with using bootstrap confidence interval for inference of the indirect effects´ statistical significance. Bootstrapping is a statistical method of simulation that aims to create a sample distribution by repeatedly resampling a single data set, in this case the indirect effects, as a mean to mimic the original sampling process.

It is, however, necessary to refer to SEM (Structural Equation Model) for estimating the indirect effects.

  The paper is evaluated highly in the following points. First, the paper extends the framework of psychological safety, strengthening its empirical support, by showing that it plays an important part in team outcomes in management teams. Second, this paper contributes to the literature of psychological safety by unfolding its little-known relationship with behavioral integration. Third, this paper suggests the mechanism how behavioral integration affects psychological safety, task performance and individual satisfaction.

Author Response

Thanks to reviewer 1 for a highly motivating review of our paper, and for showing that she/he has grasped our main aim with the study. We are not sure what reviewer 1 means by stating that it is “necessary to refer to SEM (Structural Equation Model) for estimating the indirect effects”. To clarify, we have used Andrew Hayes’ (2017) macro PROCESS for SPSS to estimate the indirect effects in the mediation model. PROCESS is designed for estimating indirect effects, and provides bootstrap confidence interval for inference of the indirect effects’ statistical significance. According to Hayes (2017) (see also Hayes, Montoya and Rockwood, 2017), using SEM to estimate indirect effects “is not necessarily better or more appropriate….For observed variable models…., it makes little difference, and your results will be unaffected by the choice” (Hayes, 2017, p. 527). We hope the reviewer agrees with us that indirect effects can be estimated with PROCESS for the observed variables in our study. 

Reviewer 2 Report

The document is well-structured and contains relevant and accurate information. The only comment I have is that the authors should give more clarity on why the two variables, "Individual satisfaction" and "Task performance," are taken to describe "Management team effectiveness."

1.The study looks into whether psychological safety indirectly impacts management team effectiveness mediated by the variable of behavioral integration.

2.The topic is relevant in the field. Organizations are looking to improve team effectiveness, and this article provides good information about the topic.

3.The main addition of the article to the subject area is the analysis of the mediating role of behavioral integration. Also, it shows that psychological safety significantly impacts management team outcomes.

4.The authors should give more clarity on why the two variables, "Individual satisfaction" and "Task performance," are taken to describe "Management team effectiveness."

5.The conclusions are consistent with the research question and the analysis of the results. They address the main question.

6.The references are appropriate. They include previous studies on the field and current studies that have been done on the topics the article integrates.

7.The figures and tables are well described.

Author Response

"Task performance," are taken to describe "Management team effectiveness."

Thanks to reviewer 2 for a highly motivating review of our paper, and for summing up what we wanted to explore and provide to the research field through our study. In the paper’s lines 138-145, and also in lines 147,164 and 170, we have tried to clarify even further why we have chosen Task performance and Individual satisfaction as the two indicators of team effectiveness in management. Our main argument is that we build on Hackman’s (2002) suggestion that team effectiveness is composed of different aspects, where one is “serving clients” (which, in an organizational setting would be equal to creating added value for the organization) which we have called “task performance”, and another aspect is “individual well-being and growth”, which we have called “individual satisfaction”. However, building on de Wit, Greer and Jehn (2012), and Marks, Mathieu and Zaccaro (2001), we deviate from Hackman’s suggestion to include a third aspect – Team viability – in the team effectiveness construct. Team viability, or “the members’ capability to work together interdependently in the future” (Hackman, 2002, p. 27) can be seen as an example of emergent states developing in the team, and team psychological safety (our main predictor variable) is a typical example of an emergent state that can be an indicator of “the members’ capability to work together interdependently in the future”. Hence, we used two of Hackman’s (2002) three dimensions as indicators of team effectiveness in management teams.